# Nebulization of Low-Dose S-Nitrosoglutathione in Diabetic Stroke Enhances Benefits of Reperfusion and Prevents Post-Thrombolysis Hemorrhage

**DOI:** 10.3390/biom11111587

**Published:** 2021-10-27

**Authors:** Syed Kashif Zaidi, Farid Ahmed, Heba Alkhatabi, Md Nasrul Hoda, Muhammad Al-Qahtani

**Affiliations:** 1Center of Excellence in Genomic Medicine Research, Faculty of Applied Medical Sciences, King Abdulaziz University, Jeddah 21589, Saudi Arabia; fahmed1@kau.edu.sa (F.A.); heba.alkhatabi@gmail.com (H.A.); mhalqahtani@kau.edu.sa (M.A.-Q.); 2Department of Neurology, Henry Ford Health System, Detroit, MI 48202, USA; nhoda1@hfhs.org

**Keywords:** diabetes, thrombosis, stroke, GSNO, IVT

## Abstract

The COVID-19 pandemic has escalated the occurrence of hypoxia including thrombotic stroke worldwide, for which nitric oxide (NO) therapy seems very promising and translatable. Therefore, various modes/routes of NO-delivery are now being tested in different clinical trials for safer, faster, and more effective interventions against ischemic insults. Intravenous (IV) infusion of S-Nitrosoglutathione (GSNO), the major endogenous molecular pool of NO, has been reported to protect against mechanical cerebral ischemia-reperfusion (IR); however, it has been never tested in any kind of “clinically” relevant thromboembolic stroke models with or without comorbidities and in combination with the thrombolytic reperfusion therapy. Moreover, “IV-effects” of higher dose of GSNO following IR-injury have been contradicted to augment stroke injury. Herein, we tested the hypothesis that nebulization of low-dose GSNO will not alter blood pressure (BP) and will mitigate stroke injury in diabetic mice via enhanced cerebral blood flow (CBF) and brain tissue oxygenation (PbtO2). GSNO-nebulization (200 μg/kgbwt) did not alter BP, but augmented the restoration of CBF, improved behavioral outcomes and reduced stroke injury. Moreover, GSNO-nebulization increased early reoxygenation of brain tissue/PbtO2 as measured at 6.5 h post-stroke following thrombolytic reperfusion, and enervated unwanted effects of late thrombolysis in diabetic stroke. We conclude that the GSNO-nebulization is safe and effective for enhancing collateral microvascular perfusion in the early hours following stroke. Hence, nebulized-GSNO therapy has the potential to be developed and translated into an affordable field therapy against ischemic events including strokes, particularly in developing countries with limited healthcare infrastructure.

## 1. Introduction

Stroke primarily happens due to a thromboembolic occlusion (~87% strokes) resulting in reduced cerebral blood flow (CBF) and decreased brain tissue oxygenation (PbtO2) [1]. More than 13 million strokes per year occur worldwide, placing stroke among the leading causes of adult mortality and disability (http://world-stroke.org, accessed on 31 August 2021) [2]. The Kingdom of Saudi Arabia (KSA) is a high-income country yet has limited healthcare infrastructures [3]; hence, the escalating rate of stroke in the KSA is frightening, and now recognized as the second major contributor to adult mortality in the nation [3]. Advancements in stroke management made faster interventions feasible in remote settings, thereby reducing the mortality but the risk of increasing number of disabled survivors remains associated [3]. Since the microvascular reperfusion and restoration of PbtO2 remains the only proven therapeutic strategy in stroke, the development of field therapies, which can be conveniently administered to the stroke victims in the primary care centers, during transportation and in the emergency departments (EDs), is essentially needed for faster interventions. Moreover, the development of adjunct therapies which will not contradict with the FDA-approved reperfusion therapies but will enhance benefits of reperfusion, is also highly encouraged [4].

Enhanced sedentary lifestyle in the KSA now has collectively invoked the uprise of various metabo-physiological disorders, increasing the risk of stroke [3,5]. Among the Arabic-speaking regions, the KSA has the most elevated rate of increase in diabetes. Of note, the American Heart Association (AHA) recently estimated ~2× higher risk of stroke among diabetic individuals compared to non-diabetics (https://www.stroke.org/en/about-stroke/stroke-risk-factors/diabetes-and-stroke-prevention, accessed on 31 August 2021). Among these metabo-physiological comorbidities including diabetes, endothelial dysfunction resulting in poor level of vasculo-protective nitric oxide (NO) remains a common pathophysiology [6]. As such, endothelial dysfunction is widely recognized as one of the significant reasons for impaired neurovascular protection and neurorestoration after stroke [7].

The translation of NO-therapy has been always an interesting therapeutic domain for vascular protection in ischemic stroke [8], yet its success has remained limited due to challenges associated with the delivery of NO [7,9]; otherwise, the risk of blood pressure (BP) lowering and subsequent hemorrhagic transformation (HT) remains warranted in stroke. Despite this, following a worldwide increase in the incidences of hypoxia and stroke due to the COVID-19 pandemic, several clinical trials have been initiated recently to test and enhance the safety and efficacy of NO-delivery via different modes and routes of administration (NCT04476992, NCT04383002, NCT04305457, NCT04337918, NCT04601077). Thus, developing the therapeutic strategy to translate an endogenous NO-carrier/donor is an appealing avenue in ischemic events including stroke [8].

S-Nitrosoglutathione (GSNO), a highly soluble small molecule major endogenous carrier/donor of NO, has shown benefits in mechanically occluded/reperfused stroke models [10,11]. GSNO-therapy also reduced embolization in human patients [12], attenuated secondary ischemia after subarachnoid hemorrhage (SAH) [13], and, of particular interest, induced hypoxic vasodilation to enhance CBF in rodents [10,11,14]. However, a risk of augmented stroke injury remains intact with the higher dose of exogenous GSNO-therapy when administered “intravenously” (IV) [15]. Therefore, prior to embarking on a possible clinical trial in future, it is critical to re-evaluate and develop alternative approaches in a preclinical setting to safely administer GSNO-therapy in stroke. Furthermore, despite promising effects in healthy animals [10,11], GSNO remains untested in any kind of clinically relevant animal models of thromboembolic stroke with comorbidity which can facilitate its preclinical validation with IV-thrombolytic (IVT) therapy, the gold standard treatment among the two FDA-approved reperfusion therapies in stroke.

In the present work, we have adopted a simple approach for the preclinical translation and validation of GSNO-therapy with or without IVT in a thrombotic stroke model in diabetic mice, a preclinical setting representing the diabetic stroke patients who are often least benefitted from the IVT-therapy due to higher risk of HT. We tested the hypothesis that nebulization of low-dose GSNO following thrombotic stroke in diabetic mice will mitigate stroke injury, improve behavioral outcomes, and enhance the benefits of thrombolytic reperfusion via improved PbtO2.

## 2. Materials and Methods

### 2.1. Animals and Experimental Approach

In-house bred male C57/B6 were used in accordance with the approved animal protocol by the ethical committee of Center of Excellence in Genomic Medicine Research (CEGMR) at King Abdulaziz University in accordance with the guidelines of National Bioethical Committee of Saudi Arabia (NACSA). All mice were subjected to streptozotocin (STZ) induced type-I diabetes for 3 weeks prior to stroke injury and GSNO nebulization. This diabetic stroke model has been validated to test therapies in a setting of post-thrombolysis risk of HT [16,17]. In Experiment I (*n* = 7 mice/set), we determined the baseline BP with vehicle control treatments followed by either IV-GSNO (1-mg per kg body weight (/kgbwt)) or GSNO-Nebulization (200 μg/kgbwt) effects on BP. In Experiment II (*n* = 12 mice/group; 2 groups; Vehicle control (+Veh) vs. GSNO treated (+GSNO)), we tested the efficacy of low-dose GSNO nebulization following thrombotic stroke on CBF, behavioral outcomes and stroke injury size (24 h and 48 h post-stroke outcomes, respectively). In Experiment III (*n* = 10 mice/group; 2 groups; +Veh vs. +GSNO), we tested GSNO-nebulization with late-IVT therapy (given at 3.5 h post-stroke) to determine PbtO2 at 6.5 h post-stroke. In Experiment IV (*n* = 20 mice/group; 2 groups; +Veh vs. +GSNO), we tested the effect of GSNO-nebulization in combination with late-IVT on behavioral outcomes, blood-brain barrier (BBB) and HT (24 h endpoint).

### 2.2. STZ-Induced Model of Type-I Diabetes

STZ-induced model of type-I diabetes has been validated for preclinical stroke and related post-thrombolysis HT-studies [16,17]. It was performed as reported elsewhere [16] but with slight modifications using 6 days STZ-injection once daily. Mice (10 ± 1-wks old; total 100 mice) were fasted overnight and injected with 100 mg/kgbwt STZ intraperitoneal (IP; Sigma-Aldrich Chemicals, St. Louis, MO, USA) on day 1. The following consecutive 5 days, STZ was given at 50 mg/kgbwt IP-dose once daily. This ensured induction of diabetes in almost all mice except 2 of them. After 3 weeks from the last STZ-injection and prior to using them for stroke experiments (13 ± 1-wks old), blood glucose level was measured with a commercial glucometer (Freestyle, Abbott Diabetes Care, Al-Ameda, CA, USA) as per the manufacturer’s protocol. Mice (*n* = 98 mice in total) with varying blood glucose levels ranging between 280–400 mg/dL were considered as diabetic, while the 2 mice with <280 mg/dL blood glucose were excluded from the study protocol.

### 2.3. BP-Measurement with Tail-Cuff

In Experiment I, multichannel MC4000 tail-cuff instrument (Hatteras Instruments, Cary, NC, USA) for BP-measurement was used as per the manufacturer’s manual and instructions. Briefly, after confirming the induction of diabetes at 3 weeks following STZ-injection, mice (*n* = 14 mice) were acclimatized with the instrument for 5 consecutive days (10 cycles of tail-cuff/day × 3 trials). On the day of measurement, mice were randomized to 2 sets (*n* = 7 mice/set); either IV-route or Nebulization-route sets. On day 6, each set was treated with their respective vehicles (either 200 μL of IV-saline as vehicle for IV-GSNO baseline control BP-data or 1 mL saline nebulized for 10 min as vehicle for GSNO-nebulization baseline control data). After 20 min of vehicle treatment, BP (systolic, diastolic, and mean arterial pressure (MAP)) was recorded and presented as the control baseline data. Next, on day 7, similar experiment was repeated for each set with either IV-GSNO (1 mg/kgbwt in 200 μL saline) or GSNO-Nebulization (200 μg/kgbwt in 1 mL saline for 10 min), and the BP was recorded and compared as the paired data with the corresponding vehicle-treated baseline data sets. Finally, both sets of mice were humanely euthanized but not included and used further in stroke experiments.

### 2.4. Modified IVT-Responsive Photothrombotic (PT)-Model of Stroke

Modified PT-stroke model which produces fibrin-rich clot was adopted from elsewhere [18], but with significant modifications such as lateral-side surgical approach without craniotomy, laser-beam source, and route of injection. Briefly, mice were anesthetized with isoflurane (1.5–2% mixed with room air) using a low-flow gas anesthesia system (Somnosuite, Kent Scientific, Torrington, CT, USA). Mice were then transferred to a stereotaxic frame integrated with an angle device (Stoelting Co, Wood Dale, IL, USA) and a 532 nm laser source (4.5 mW; beam diameter 2.5–3 mm; ThorLabs, Newton, NJ, USA). The head was rotated to face right side up, hair was shaved, the temporal muscles under the skin were blunt dissected and the skull-bone above the zygomatic arc was exposed. An admixture of thrombin and Rose Bengal (RB) dye (80 U/kgbwt and 50 mg/kgbwt, respectively, in 200 μL of 1× sterile saline; Sigma) was then IV-injected via the tail vein. Using the laser light, the distal trunk of middle cerebral artery (MCA) was visualized underneath the bone and the laser was lowered down to the skull bone, illuminated, and exposed for 15 min. Thrombotic occlusion of the distal lateral trunk of MCA was confirmed with the LDF (>70% drop in focal CBF). Finally, wound was closed carefully using 3M VetBond tissue glue (Santa Cruz Biotechnology, Santa Cruz, CA, USA). Mice were then transferred to a clean warm cage for recovery (4-mice/cage) and provided food and water freely.

### 2.5. Treatments and Therapies after Stroke

From each block (4-stroked mice/cage), mice were block-randomized to different treatments as needed for two groups in each experiment. In all 3 experiments, Aeroneb lab nebulizer unit small VMD integrated to a nose-cone animal holder and Somnosuite (Kent Scientific), was used for nebulization (1 mL volume for 10 min) in conscious mice as per the vendor’s instructions. Either vehicle (1× sterile saline, +Veh group) or GSNO (Sigma; 100 µg/kgbwt in saline × 2 doses at 2 h and 6 h post-stroke) were nebulized as needed for different groups. For IVT (+IVT groups in Experiments III and IV), recombinant tissue plasminogen activator (rt-PA; Altepase, Genentech, San Francisco, CA, USA) was used as reported earlier by us [19]; however, since this PT-stroke is a model of smaller clot formation and injury, we used rt-PA at a dose of 5 mg/kgbwt. In our hands, this dose of IVT beyond 3 h post-stroke does not cause unexpected higher mortality, as often seen in diabetic stroke, but results in effective thrombolysis and shows signs of HT, which serves the purpose of this study.

### 2.6. Laser Doppler Flowmetry (LDF) and Laser Speckle Contrast Analysis (LASCA) Imaging

LDF (Perimed (Sweden) Inc., Las Vegas, NV, USA) for focal CBF in Experiment II–IV, and LASCA imaging (RFLSI-III, RWD Life Sciences, San Diego, CA) in Experiment II for global relative CBF (% change) were performed as reported by us [19,20]. Briefly, an LDF-probe (PH407, Perimed) was inserted into the LDF probe holder (PH07-6, Perimed; pre-fixed at the stereotaxic coordinates, AP: 2 mm and ML: 3 mm with respect to bregma). The LDF perfusion signal (in arbitrary unit, AU) was recorded prior to initiating the induction of stroke and 10 min after the completion of laser exposure to confirm the induction of stroke. In brief, for LASCA imaging at 24 h post-stroke, anesthetized mice were maintained at 37 ± 0.5 °C body temperature, and the recording was performed on an exposed clean skull. The acquired perfusion videos were analyzed for % CBF changes in the ipsilateral ischemic hemisphere normalized to the uninjured contralateral hemisphere.

### 2.7. Behavioral Tests

For all 3 behavioral tests, mice were pre-trained for 3 days (3 trials/day) prior to the stroke surgery. These tests have been recommended to determine subtle changes in behavioral outcomes following stroke models of smaller injury [19,20,21,22,23]. In Experiment II, at 48 h post-stroke and prior to sacrifice, the motor balance function of mice was assessed with the balance beam walk test (BWT). Briefly, pre-trained stroke mice were individually tested to walk on a 100 cm long beam with 3 trials × 5 min interval between each trial. BWT performance of each mouse was quantified by measuring the time taken (in seconds, s) by the mouse to traverse the beam, and the mean value of 3 trials was calculated and presented as the BWT outcome measure [21,22]. In Experiment IV, adhesive tape test (ATT) as the somatosensory functional outcome at 24 h post-stroke was performed as reported by us [19], prior to sacrificing both sets of groups for the estimation of Evan’s blue (EB) content for blood-brain barrier (BBB) leakage and hemoglobin (Hb) content as the marker of HT. As reported by us [19], higher time (in second, s) taken by the mouse to remove the tactile stimuli (adhesive tape) from the paw indicates a poorer outcome and vice versa in this ATT method. In addition, hanging wire test (HWT) for the muscular strength and coordination in Experiment IV also was performed as reported elsewhere [21]. Briefly, pre-trained stroked mice were gently encouraged to hold a hanging wire with both forelimbs. The fall latency (in second, s) was recorded for 3 trials × 5 min interval between trials for each mouse, and the mean value was calculated to present as the outcome measure of the HWT.

### 2.8. 2,3,5-Triphenyltetrazolium Chloride (TTC) Staining for Infarct Volume Analysis

TTC-stain differentiates between metabolically active (live or penumbra) and inactive (dead or core) tissues after stroke as reported by us [19]. In Experiment II, it was performed to measure the infarct/stroke injury size at 48 h post-stroke [19].

### 2.9. Measurement of Brain Tissue Oxygen (PbtO2) with Fluorescence Quenching Optic Probes

Optical measurement of PbtO2 was performed at 6.5 h post-stroke and after the 2nd dose nebulization of GSNO (terminal Experiment III). The optical measurement technique for O2 carries many advantages over the classical electrochemical method, such as that no oxygen consumption occurs during the optical method. The polymer membrane of the probe, loaded with a luminescent dye, glows in the absence of oxygen when exposed to the light passing through the probe. In response to collision with available oxygen molecules, luminescence quenching occurs; thus, higher oxygen concentration results in the lesser back transmittance of the light to the detector and vice versa. Briefly, for in vivo optical recording of PbtO2, an optic fiber (OXR50, tip diameter 50–70 µm (patented technology of Pyro Science GmbH, Aachen-Germany); Ohio Lumex, Solon, OH, USA) was inserted into the brain (stereotaxic coordinates, AP: 0.5 mm and ML: 1.5 mm with respect to bregma) about 1.5 mm deep (dorso-ventral, DL) and immediately retracted back ~0.5 mm to create a void space. The pre-calibrated signal output (in mm of Hg) was stabilized for ~2–4 min and finally recorded and averaged for the next 3 min duration.

### 2.10. Measurement of BBB-Leakage and Hemorrhagic Transformation (HT)

Changes in BBB-integrity and HT following late-IVT therapy (3.5 h post-stroke) and their modulation with GSNO-nebulization as an adjunct therapy was performed as reported earlier by us with certain modifications [19,24]. Briefly, in Experiment IV, 100 μL of EB-dye (Sigma; 5% solution in sterile 1× saline) was IV-injected at 6 h post-stroke. Mice were sacrificed following neurobehavioral tests (ATT and HWT) at 24 h post-stroke. Mice were perfused with chilled 1× PBS, ipsilateral hemispheres were harvested, weighed, homogenized in 650 μL of 1× PBS and sonicated. An equal volume of 50% trichloroacetic acid (TCA, Sigma) was added and incubated overnight in the dark at 4 °C. Finally, samples were centrifuged (30 min, 15,000 rpm, 4 °C), supernatants were collected, the absorbance of EB-dye was measured at 610 nm and quantified according to a standard curve. The results were presented as μg of EB/g of tissue. For HT volume quantitation, Hb-content (in µL) was estimated as reported by us and others [20,25]. Briefly, 20 μL of clear supernatant obtained from the ipsilateral tissue homogenate was mixed with 80 μL of Drabkin’s Reagent (Sigma) in triplicates in a 96-well plate. The absorbance after 15 min was read at 540 nm. Hb-content in the brain samples were calculated from a standard curve using the mean absorbance for each sample.

### 2.11. Data Collection and Statistical Analysis

*Paired t-test* was used to compare the baseline vehicle treated control BP data to that of the corresponding GSNO-treated data in Experiment I (*n* = 7). For stroke experiments (II–IV), based on our prior experiences for changes in infarct size as the primary outcomes at 48 h, we anticipated at least 20% reduction in the infarct volume with up to 20% standard deviation (SD). Therefore, we set the sample size to achieve at least 80% power at 5% level of significance to detect main effects. In each experiment, mice were block randomized in blocks of 4 mice/cage for the two groups. Data collection and analyses were blinded to the investigators. Quantitative data are presented as mean ± SD, statistical analysis was performed with at least *n* ≥ 8 and a *p*-value < 0.05 was considered statistically significant. *Unpaired t-test* was performed to determine the statistical significance, and wherever needed, a log transformation was used prior to statistical analysis to stabilize the variance across the groups.

## 3. Results

### 3.1. IV-GSNO But Not the GSNO-Nebulization Causes BP-Lowering

As reported by others in normal mice [15], IV-GSNO (1 mg/kgbwt) following thrombotic stroke in diabetic mice also did not confer protection in our hands, but augmented the HT. Since the GSNO is a major endogenous NO-carrier/donor, we first sought to determine whether a change in the route of administration of GSNO will abolish its BP-lowering effect. As expected, a higher dose of IV-GSNO at 1 mg/kgbwt significantly lowered the BP as compared to the baseline control (*p* < 0.0001; Figure 1A); however, a low-dose GSNO-nebulization (200 µg/kgbwt) did not alter the BP significantly (Figure 1B). Therefore, in stroke experiments, we used low-dose GSNO via nebulization (100 µg/kgbwt × twice at 2 h and 6 h post-stroke).

### 3.2. Low-Dose of GSNO-Nebulization Improves CBF and Mitigates Thrombotic Stroke Injury in Diabetic Mice

IV-treatment with higher dose GSNO (1 mg/kgbwt) protects against mechanical occlusion-reperfusion injury in normal animals [10,11] but also has been contradicted [15], likely due to adverse effects related to BP-lowering. Therefore, next we determined whether the nebulization of low-dose GSNO following a spontaneously reperfusing thrombotic stroke in diabetic mice (a clinically relevant model) is protective (Figure 2 and Figure 3). GSNO nebulization, initiated as late as 2 h post-stroke, showed enhanced CBF at 24 h post-stroke in +GSNO group as compared to the +Veh control (79.9 ± 13.9% vs. 59.3 ± 11.9%, respectively, *p* < 0.0008; Figure 2A,B). Therefore, as expected, the behavioral outcome as tested with BWT at 48 h post-stroke also was improved in +GSNO group vs. the +Veh control (28.2 ± 3.2 s vs. 22.5 ± 4.2 s, respectively, *p* < 0.0011; Figure 3A). Moreover, we found that the infarct volume at 48 h post-stroke also was significantly reduced in the +GSNO group vs. +Veh control mice (39.6 ± 3.9 mm^3^ vs. 45.5 ± 5.7 mm^3^, respectively, *p* < 0.0074; Figure 3B,C). Hence, our data is a proof-of-concept that nebulization of low-dose GSNO is safe and protective in a non-reperfused and/or spontaneously reperfusing thrombotic stroke model with comorbidity.

### 3.3. Nebulization of Low-Dose GSNO Enhances Post-Thrombolysis Brain Tissue Oxygenation (PbtO2) in Diabetic Mice

Thrombolytic recanalization of an occluded artery may not necessarily translate into an optimal microvascular perfusion and enhanced PbtO2, the master requisite to treat an ischemic brain. Since GSNO causes hypoxic vasodilation resulting in enhanced PbtO2, we next sought whether GSNO-therapy, as an adjunct to IVT, has the potential to improve PbtO2 following thrombolysis in stroke. As evident from Figure 4A,B “+GSNO with IVT” enhanced the PbtO2 at 6.5 h post-stroke significantly as compared to “+Veh with IVT” (50.8 ± 9.3 mm Hg vs. 41.7 ± 8.8 mm Hg, respectively, *p* < 0.0379). Our data demonstrate that late-IVT therapy in diabetic stroke does not optimally restore PbtO2; however, a combination with GSNO further enhances the restoration of PbtO2 following late thrombolysis in diabetic mice.

### 3.4. Nebulized GSNO in Combination with Late Thrombolysis Augments Behavioral Outcomes

GSNO alone has been reported to improve behavioral outcomes following mechanical occlusion-reperfusion in rodents; however, the potential benefits of GSNO-therapy on behavioral outcomes remain unknown in a setting of thrombotic occlusion and in combination with late-IVT therapy. Therefore, in a larger set of mice (*n* = 17–20 mice/group), we investigated the benefits of GSNO in combination with late-IVT therapy in improving behavioral outcomes (Figure 5A,B). As evident, “+GSNO with IVT” compared to “+Veh with IVT” significantly enhanced the somatosensory functional outcomes as tested with ATT (36.8 ± 8.8 s vs. 46.1 ± 9.5 s, *p* < 0.0042), and muscular strength and coordination to grip (motor coordination) as determined with HWT (40.45 ± 8.2 s vs. 34.5 46.1 ± 5.9 s, *p* < 0.0164). Thus, our data demonstrate that nebulized GSNO supplement with late-IVT therapy is beneficial in improving functional outcomes.

### 3.5. Low Dose GSNO Nebulization Attenuates Detrimental Effects of Late Thrombolysis in Diabetic Mice

The FDA approves and recommends the use of IVT-therapy in stroke within 3 h post-ictus beyond which IVT could augment the BBB-leakage and HT itself. Moreover, pre-existing diabetes exacerbates the risk of BBB-disruption and HT after stroke [16,17], particularly in the setting of late-IVT treatment, thereby shortening the safer thrombolytic window and limiting the benefits of IVT-therapy in diabetic stroke. Therefore, in view of the critical translational relevance, we next evaluated whether GSNO would abolish or attenuate BBB-disruption and HT following late-IVT in diabetic mice (Figure 6A–C). As evident from the representative images (Figure 6A), late-IVT at 3.5 h post-stroke not only augmented the BBB-leakage as determined by more intense appearance of EB-stain in +Veh control group but also increased the HT (darker patches of blood under the EB-stain). On the other hand, as it can be seen, this was collectively attenuated in the +GSNO group with IVT. Furthermore, quantification of EB- and Hb-contents demonstrated that “+GSNO with IVT” significantly prevented the EB-content (*p* < 0.0363) and Hb-content (*p* < 0.0006) as compared to the “+Veh with IVT” controls (Figure 6B,C), thus evidencing that low-dose GSNO nebulization attenuates the detrimental effects of late-IVT on the BBB integrity and reduces the risk of HT in diabetic mice.

## 4. Discussion

To the best our knowledge, this is the first report demonstrating the efficacy of GSNO, an endogenous major carrier/donor of NO, in any kind of thromboembolic stroke model with comorbidity and in combination with IVT. Our key findings are that GSNO-therapy alone in thrombotic stroke improves neuro-functional outcomes, likely via enhanced CBF. Moreover, our novel finding demonstrates, for the first time in diabetic mice subjected to thrombotic stroke, that GSNO enhances the benefits of thrombolytic reperfusion by augmenting brain tissue reoxygenation/PbtO2 and enervates the detrimental effects of late thrombolysis. In view of the escalated risk of stroke injury/HT in diabetic subjects and an increased interest in innovative NO-delivery in COVID-related hypoxia-ischemia, our finding in this report bears critical translational relevance.

Despite several reports related to GSNO-therapy in animal models of mechanically occluded/reperfused stroke models [10,11], its efficacy following IV-injection of higher dose (1 mg/kgbwt) was contradicted to augment stroke injury [15]. Our initial studies in a comorbid/diabetic model of thrombotic stroke yielded similar results, showing frequent HT when given IV-GSNO at 1 mg/kgbwt, which compelled us to re-evaluate the dose and route of administration of GSNO. We note that our study was initiated in a comorbid thrombotic stroke model instead of mechanical IR-model of stroke in normal mice; our goal here is not to contradict prior reports but to fine-tune the route of administration and dose of GSNO-therapy for safer clinical translation in future. Since GSNO reduces embolization in human patients [12], attenuates ischemia [13] and induces hypoxic vasodilation [14], its clinical translation as a field therapy to improve collateral microcirculation and PbtO2 in the early hours, particularly in view of the frequent thrombotic strokes due to COVID-19, is promising.

The benefits of BP-lowering in stroke remains debated, although reduction of systolic BP prior to IVT-therapy could be beneficial in subjects with extreme hypertension [26]. On the other hand, significant reduction of BP, even to the lower levels of the hypertensive range, could be detrimental in stroke patients [27]. As evident from our data (Figure 1), a higher dose of IV-GSNO had a significant effect on BP-lowering (Figure 1A) which likely resulted in frequent HT, as seen by us (Figure 6), and exacerbated stroke injury, as reported by others [15]. On the other hand, low-dose nebulization of GSNO remains well-tolerated and did not alter the BP (Figure 1B). Since nebulization, as the route of administration, bears higher translational applicability in clinical stroke settings and allows enhanced control for safer drug delivery, we found it encouraging to test in thrombotic stroke with diabetes. When tested, low-dose GSNO-nebulization alone enhanced the CBF, improved behavioral outcomes and attenuated the progression of stroke injury in diabetic mice without altering the BP (Figure 1, Figure 2 and Figure 3). Moreover, in combination with the thrombolytic reperfusion therapy, GSNO-nebulization did not cause any adverse effects but diminished the unwanted effects of late-IVT in diabetic mice, likely via enhanced re-oxygenation in early hours (Figure 4, Figure 5 and Figure 6), which eventually translated into improved behavioral outcomes. These findings indicate that the nebulization of GSNO hits the key therapeutic goals required to treat a stroke in early hours.

Successful reperfusion in major vessels and restoration of CBF therein do not necessarily guarantee the restoration of microvascular perfusion, eventually responsible for reinstating the oxygenation of brain tissue [28,29,30]. Therefore, the development of affordable field therapies in stroke, which will be conveniently accessible to paramedics and safer to use at primary care centers, during transportation in ambulance and in EDs, is desperately needed in developing countries such as the KSA, with limited healthcare infrastructure. Most importantly, if field therapies are effective at enhancing collateral microcirculation and PbtO2 in the early hours after stroke, this will have the benefits of reducing mortalities, the severity of long-term disabilities and associated healthcare costs. Therefore, in summary, our findings in this report draw attention to further validate GSNO-therapy in different models of thromboembolic stroke with and without comorbidities in aged animals of both sexes to better represent a clinical stroke scenario. Moreover, NO-therapy is being actively tested worldwide in severe COVID-19 patients in need of exogenous oxygen support (NICOR trial: NCT04476992; and NCT04383002), and different modes of NO-delivery in mild-moderate COVID-19 patients, such as NO-inhalation (NCT04305457), NO-releasing solutions (NOCOVID trial and NCT04337918) and NO-lozenges (NCT04601077) etc. Since COVID-19 patients remain at higher risk of thrombotic stroke due to cytokine storm and resultant acute respiratory disorder syndrome (ARDS), future preclinical testing of GSNO as a NO-therapy is also encouraged in preclinical models of COVID-19 and/or ARDS accompanying stroke.

## Figures and Tables

**Figure 1 biomolecules-11-01587-f001:**
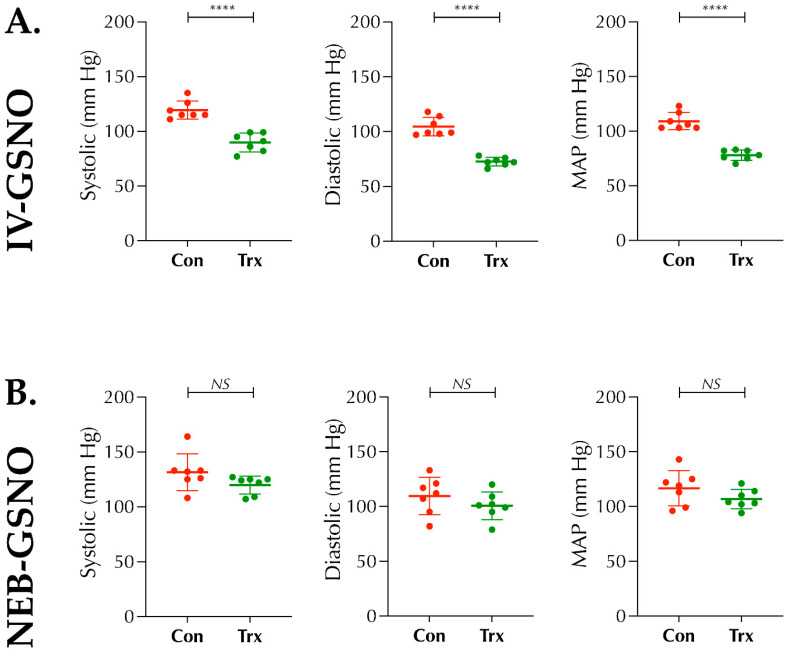
Effect of the route of administration of exogenous s-nitrosoglutathione (GSNO) on blood pressure (BP): Diabetic male mice (C57/B6 background, 14 ± 1-wks. old) were acclimatized with the tail-cuff instrument and subjected to BP-measurements with or without Veh/IV-GSNO or Veh/GSNO-Nebulization. BP-measurement was performed 20 min later. (**A**) Significant effect of IV-GSNO treatment (1 mg/kgbwt) was found on BP-lowering compared to their baseline control (****, *p* = 0.0001), and (**B**) No significant (NS) alteration was found with GSNO-nebulization (200 µg/kgbwt) compared to their baseline control (NS; *p* = 0.174 for Mean Arterial Pressure, MAP). Data are presented as Mean ± SD (*n* = 7 mice/group). Statistical significance was determined at *p* < 0.05 and using a *paired t-test*.

**Figure 2 biomolecules-11-01587-f002:**
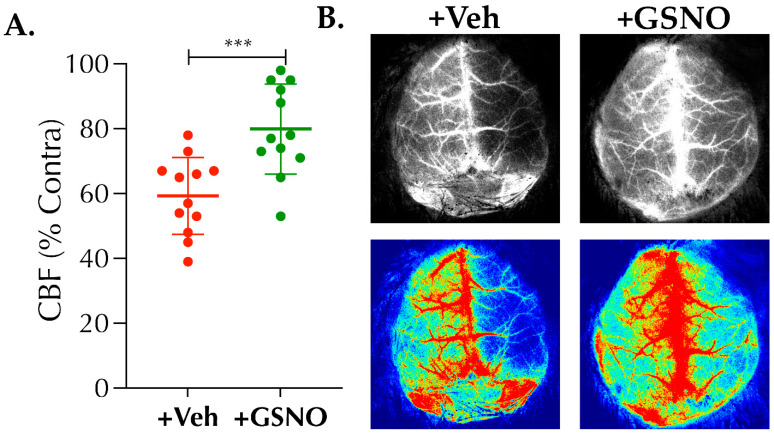
Nebulization of s-nitrosoglutathione (GSNO) following thrombotic stroke in diabetic mice enhances cerebral blood flow (CBF). Diabetic male mice (C57/B6 background, 14 ± 1-wks. old) were subjected to thrombotic (photothrombotic, PT) stroke. Mice were randomized and treated with either saline as vehicle (+Veh control group) or GSNO (+GSNO treated group; 100 µg/kgbwt × 2 doses at 2 h and 6 h post-stroke) nebulized in a total volume of 1 mL over a period of 10 min. Relative CBF (% contra) was measured at 24 h post-stroke using a laser speckle contrast imager (LSCI). (**A**) GSNO nebulization in +GSNO group significantly enhanced the % CBF change as compared to the +Veh group (***, *p* = 0.0008). (**B**) Representative laser speckle maps (top greyscale images) and pseudo color images (bottom-colored) showing changes in CBF in +Veh and +GSNO groups as indicated. Data are presented as Mean ± SD (*n* = 12 mice/group). Statistical significance was determined at *p* < 0.05 and using an *unpaired t-test*. No mortalities were observed by 24 h post-stroke at the time of LSCI procedure as rarely seen in this mini-infarct model of PT-stroke.

**Figure 3 biomolecules-11-01587-f003:**
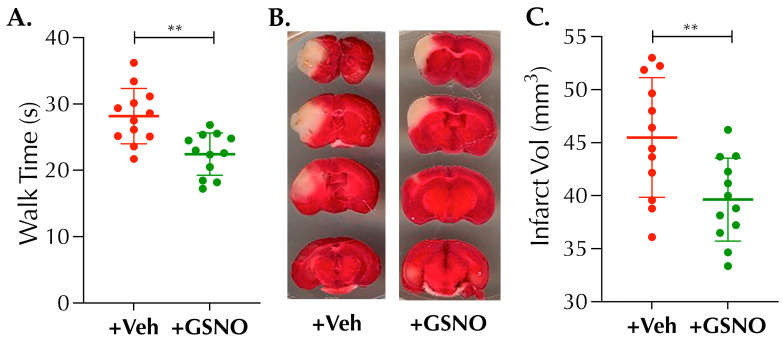
Nebulization of GSNO improves behavioral outcomes and attenuates thrombotic stroke injury in diabetic mice. Diabetic male mice (C57/B6 background, 14 ± 1-wks. old) were subjected to PT-stroke and treated with Veh or GSNO as described in Figure 1. Behavioral assessment on beam-walk test (BWT) and absolute infarct volume (in mm^3^) /stroke injury analyses were performed at 48 h post-stroke. (**A**) GSNO nebulization (100 µg/kgbwt × 2 doses at 2 h and 6 h post-stroke) significantly improved the behavioral outcomes in +GSNO group as compared to the +Veh group (**, *p* = 0.0011). Moreover, (**B**,**C**) GSNO nebulization was also significantly effective in preventing the infarct progression in +GSNO group as compared to +Veh controls (**, *p* = 0.0074). Data are presented as Mean ± SD (*n* = 12 mice/group). Statistical significance was determined at *p* < 0.05 and using an *unpaired t-test*. No mortalities were observed by the end of this experiment at 48 h, as rarely seen in this mini-infarct model of PT-stroke.

**Figure 4 biomolecules-11-01587-f004:**
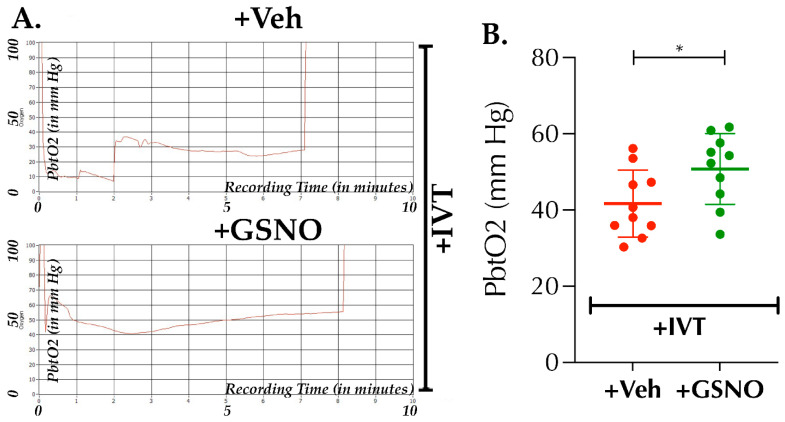
Nebulization of GSNO augments post-thrombolysis brain tissue oxygenation (PbtO2) at 6.5 h after stroke in diabetic mice. Diabetic male mice (C57/B6 background, 14 ± 1-wks. old) were subjected to PT-stroke and treated with either Veh or GSNO (100 µg/kgbwt × 2 doses at 2 h and 6 h post-stroke) as described in Figure 1. In addition, all mice from both groups also were treated with the FDA-approved IVT-therapy, recombinant tissue plasminogen activator (rt-PA; 5 mg/kg bwt), given at 3.5 h post-stroke. PbtO2 was measured at 6.5 h post-stroke, i.e., 3 h post-thrombolysis and 20 min after the completion of 2nd dose nebulization of GSNO. (**A**) Representative real-time traces of fluorescence quenching, recorded for the measurement of PbtO2 for Veh + IVT and GSNO + IVT groups, respectively. (**B**) GSNO nebulization in combination with late-IVT therapy significantly enhanced the PbtO2-value as compared to the late-IVT alone controls (*, *p* = 0.0379). Data are presented as Mean ± SD (*n* = 10 mice/group). Statistical significance was determined at *p* < 0.05 and using an *unpaired t-test*. This was a terminal experiment at 6.5 h post-stroke following PbtO2 measurement, and no mortalities were observed until sacrificed.

**Figure 5 biomolecules-11-01587-f005:**
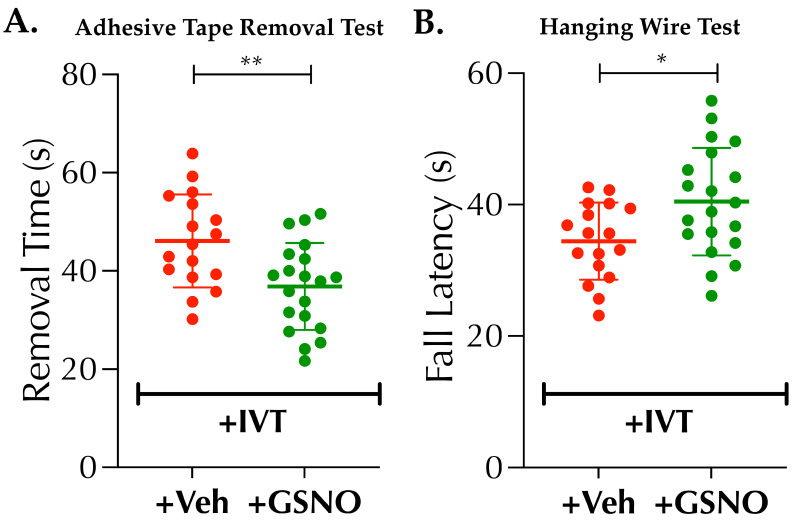
Nebulization of GSNO improves post-thrombolysis behavioral outcomes in diabetic mice subjected to thrombotic stroke. Diabetic male mice (C57/B6 background, 14 ± 1-wks. old) were subjected to PT-stroke and treated with either Veh or GSNO (100 µg/kgbwt × 2 doses at 2 h and 6 h post-stroke) as described in Figure 1, followed by thrombolysis in all mice as mentioned in Figure 4. At 6 h post-stroke, mice (*n* = 10 mice/group) were infused with the tracer dye, Evan’s blue (EB), for the assessment of BBB-leakage and integrity at 24 h post-stroke; and the remaining (*n* = 10 mice/group) were left without EB-infusion for the estimation of hemoglobin (Hb) content as the marker of post-thrombolysis HT at 24 h post-stroke. All mice from above 2 experiments for EB- and HT-contents were subjected to both, adhesive tape test (ATT) for somatosensory functional outcomes and hanging wire test (HWT) for muscular strength and coordination, prior to sacrifice at 24 h post-stroke. (**A**) GSNO-therapy in combination with late-IVT (GSNO + IVT group) significantly improved the somatosensory functions as compared to late-IVT alone (Veh + IVT) controls (**, *p* = 0.0042). (**B**) GSNO-therapy in combination with IVT-therapy (GSNO + IVT group) also significantly augmented the muscular strength, coordination and use functions as compared to Veh + IVT controls (*, *p* = 0.0164). Data are presented as Mean ± SD (*n* = 17–20 mice/group). Statistical significance was determined at *p* < 0.05 and using an unpaired t-test. Following behavioral outcome measures, all surviving mice of both sets were euthanized for the quantification of EB- and Hb-contents.

**Figure 6 biomolecules-11-01587-f006:**
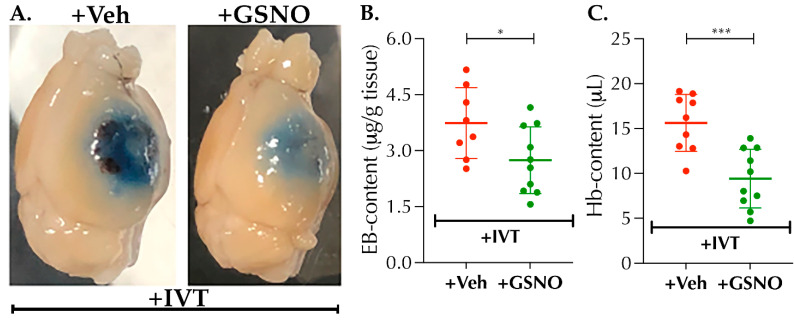
Nebulization of GSNO attenuates post-thrombolysis blood-brain barrier (BBB) leakage and hemorrhagic transformation (HT) in diabetic mice subjected to thrombotic stroke. Diabetic male mice (C57/B6 background, 14 ± 1-wks. old) were subjected to stroke, treated, and followed up to 24 h as described in Figure 5. (**A**) Representative images of the whole brain showing exacerbated EB/ BBB-leakage and prominent HT in the Veh + IVT group in response to late-IVT therapy at 3.5 h (5 mg/kgbwt), which was collectively reduced in the GSNO + IVT group. (**B**) Nebulization of GSNO (100 µg/kgbwt × 2 doses at 2 h and 6 h post-stroke) in combination with late-IVT (GSNO + IVT group) significantly reduced the BBB-leakage as compared to late-IVT alone (Veh + IVT group) controls (*, *p* = 0.0363, *n* = 8–10 mice/group). **(C)** Moreover, HT also was significantly attenuated in the GSNO + IVT group as compared to the Veh + IVT controls (***, *p* = 0.0006, *n* = 9–10 mice/group). Statistical significance was determined at *p* < 0.05 and using an *unpaired t-test*. No mortalities were observed in both sets of GSNO + IVT group; however, Veh + IVT group showed overall 3 mortalities out of 20 mice likely due to late IVT-therapy and the resulting significant HT in Veh + IVT group.

## Data Availability

Not Applicable.

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
