# Peer review of "Nebulization of Low-Dose S-Nitrosoglutathione in Diabetic Stroke Enhances Benefits of Reperfusion and Prevents Post-Thrombolysis Hemorrhage"

_biomolecules, 2021, doi:10.3390/biom11111587_

Round 1

Reviewer 1 Report

The present study shows that "Nebulization of Low Dose S-Nitrosoglutathione in Diabetic Stroke Enhances Benefits of Reperfusion and Prevents Post-Thrombolysis Hemorrhage" is an interesting and innovative research in the area of Stroke. This article stems from the recent excellent US patent published by the authors describing beautifully the connection between diabetic stroke and nebulization of GSNO. Hence, this study is novel, and the authors have tried to establish the role of GSNO in mediating post thrombolysis hemorrhage. The study has been done quite well and various aspects have been also taken into consideration. Data generated in this study are convincing and observed novel findings. The article is very well written in light of published articles, however, in my opinion the paper has some typo, and these should be addressed before the final decision is made.

Minor comments:

  1. Language is quite easily understandable. However, in few cases, there are few mistakes in terms of grammar and writing mentioned below:

Page 1: Line 10 (has been reported), 27 (per year) and 44 (estimated at least).

Page 2: Line 79.

Page 4: Line 174 (add reference…as reported by us..), line 184 (delete others).

Page 5: Line 212 {(content (in μl)}

Page 10: Line 367 (delete here from “date not shown here”).

  1. There are few references that need to be edited or replaced by mentioned PMIDs.

Page 2: Line 91 PMIDs: 22052516; 25307224

Page 4: Line 164 PMID: 20642841

            Line 178: PMID: 21756996

Page 5: Line 232: PMID: 18796513

Page 6: Line 244: Khan, 2006 # 235; Khan, 2005 # 236

            Line 245: Killic, 2008 # 216

Page 11: Line 381: Killic, 2008 # 216

  1. Figure 4A graph is the real-time output of the machine. If possible please provide a clear image, the scales mentioned in the graph are not clear.

Author Response

Dear Editor,

Please find our attached revised manuscript titled "Nebulization of Low Dose S-Nitrosoglutathione in Diabetic Stroke Enhances Benefits of Reperfusion and Prevents Post-Thrombolysis Hemorrhage". We apologize for the typographical and grammatical errors. We gave ample time again and have now thoroughly revised the manuscript including figures and references as kindly suggested by the learned reviewer, we also have enhanced the resolution of Figure #4 as advised by the Reviewer. As such there are no significant additions but we have cited few essential references and corrected any discrepancies due to typos. We hope that our reviewer and you will find our edits satisfactory. We thank you and our reviewer for your time and constructive comments which have certainly helped to enhance the quality of the manuscript. We have tabulated our responses actions to the reviewer’s comments.

Thank you very much again for your assistance, and re-consideration of our manuscript.

Regards, 

Syed Kashif Zaidi, Ph.D.

Reviewer 2 Report

In this study, Zaidi et al investigate the protective effects of low dose nebulization of S-Nitrosoglutathione (GSNO), a nitric oxide donor, in diabetic mice following photothrombotic stroke. The authors found that intravenous injection with high dose of GSNO (1 mg/kg bwt) significantly decreased blood pressure in streptozotocin (STZ)-induced diabetic mice, while nebulization with low dose of GSNO (200 µg/kg bwt) did not alter blood pressure in the diabetic mice when compared to their baseline control. Two doses of GSNO nebulization (100 µg/kg bwt) at 2h and 6h post-stroke significantly reduced brain infarct volume and improved motor function at 48h in diabetic mice following photothrombotic stroke, and these protective effects were associated with a dramatic increase in cerebral blood flow (CBF) compared to vehicle group. Further studies demonstrated that low dose of GSNO nebulization (100 µg/kg bwt) in combination with recombinant tissue plasminogen activator (rt-PA; 5 mg/kg bwt) attenuated post-thrombolysis neurobehavioral impairment, blood-brain barrier (BBB) leakage and hemorrhagic transformation in diabetic mice subjected to photothrombotic stroke, and these effects were associated with an augment of brain tissue oxygenation level. The authors present interesting findings of GSNO in a clinically relevant thrombotic stroke model using diabetic mice and in combination with thrombolysis. However, there are some omissions, limitations and errors that need to be addressed before this can be considered further for publication.

Major comments:

1) Many risk factors for stroke have been reported including hypertension, diabetes, smoking, and others. It will strengthen the manuscript with adding one paragraph in the Discussion section to demonstrate why stroke mice in comorbidity with diabetes were used in this study.

2) It is strongly suggested to move Figure 1 from the page 11 to the page 5 following the section “3.1 IV-GSNO but Not the GSNO-Nebulization Causes BP-Lowering”.

3) It will be helpful to easily understand the Figure 5 if the authors can add subtitle “Adhesive Removal Test” to panel 5A and use the “Time to remove (Seconds)” instead of “ATT (Time in Sec)”. Also, add subtitle “Hanging Wire Test” to panel 5B and use the “Fall latency (Seconds)” instead of “HWT (Time in Sec)”.

Minor comments:

1) In line 17 on page 1 in the Abstract section, it seems a word such as “increased” was missing in the sentence “Moreover, GSNO-nebulization the reoxygenation……”.

2) In the section 3.1 on page 5, text of Fig.1A and Fig.1B should be included in this paragraph.

3) Two doses of GSNO were nebulized at 2h and 6h post-stroke as demonstrated in the Material and Methods section 2.5 on page 3, but the description in the line 261 for the Figure 2 legend on page 6 was at 2h and 5h post-stroke, which was not consistent.

4) In panel 4A of Figure 4, it is suggested to replace the image in a higher resolution since it is not readable.

Author Response

Dear Reviewer,

Please find our attached revised manuscript titled "Nebulization of Low Dose S-Nitrosoglutathione in Diabetic Stroke Enhances Benefits of Reperfusion and Prevents Post-Thrombolysis Hemorrhage". We apologize for the typographical and grammatical errors. We gave ample time again and have now thoroughly revised the manuscript including figures and references as kindly suggested by the learned reviewer. As suggested by the Reviewer, we paid special attention to edit the figure legends of Figure #5 and added few relevant sentences related to diabetes in the discussion section. We also have enhanced the resolution of Figure #4 as advised by the Reviewer. We hope that our reviewer and you will find our edits satisfactory. We thank you and our reviewer for your time and constructive comments which have certainly helped to enhance the quality of the manuscript. We have tabulated our responses actions to the reviewer’s comments.

Thank you 

Regards, 

Syed Kashif Zaidi, Ph.D.
